# The role of obesity and Type 2 diabetes in lung health: A systematic review (2024)

Roanne Lecky[1], Svitlana Grogan[1], Priyank Shukla[1], Sarah Atkinson[2], Paula L. McClean[1], Catriona Kelly[1]*

**1** Personalised Medicine Centre, School of Medicine, Ulster University, C-TRIC Building, Altnagelvin Hospital Campus, Derry/Londonderry, Northern Ireland, United Kingdom, **2** Centre for Genomics Medicine, School of Biomedical Sciences, Ulster University, Coleraine, Northern Ireland, United Kingdom

* c.kelly@ulster.ac.uk

## Abstract

### Introduction

Type 2 diabetes is associated with mild airways restriction, yet obstructive lung conditions are prevalent in people with diabetes. Obesity is a confounding factor and has been reported to be both protective and to enhance risk of lung disease independent of hyperglycaemia. The aim of this systematic review was to evaluate how Type 2 diabetes and obesity affect lung function measurements in people with and without chronic obstructive pulmonary disease (COPD) and asthma.

### Methods

Ovid MEDLINE and Embase databases were methodically searched for studies published between 2011–2024. Ninety-three studies were included, with 35,891 participants. Included studies had data on Type 2 diabetes and/or obesity and forced expiratory volume in one second (FEV1) and/or forced vital capacity (FVC). All studies were assessed for quality (Newcastle-Ottawa scale) or risk of bias (Cochrane methodology). Data was extracted as combined means and standard deviation, and significance tested by Kruskal-Wallis. Multiple linear regression was conducted to account for the impact of age, BMI, Type 2 diabetes and geographical region.

### Results

Those with Type 2 diabetes without a lung disease had mild airway restriction. However, outcomes for those with asthma and COPD in the presence of Type 2 diabetes were largely comparable to those who had either condition in the absence of Type 2 diabetes. A Type 2 diabetes diagnosis and being in the geographical region of Asia were significantly associated with decreased FEV1 and FVC, but obesity was not. The study is limited by the large number of cross-sectional studies using single time points from which conclusions on causality cannot be drawn.

**Data availability statement:** All data underlying the results is available in the manuscript and supporting information materials.

**Funding:** R.L. and S.G are supported by PhD Studentships from the Department for the Economy, Northern Ireland. The funders had no role in study design, data collection and analysis, decision to publish, or preparation of the manuscript.

**Competing interests:** The authors have declared that no competing interests exist.

## Conclusion

Type 2 diabetes is independently associated with airways restriction suggesting that monitoring of lung function following a diabetes diagnosis may be warranted.

---

## Introduction

Microvascular and macrovascular complications significantly contribute to the morbidity and mortality associated with Type 2 diabetes. Traditionally, the lungs have not been regarded as an end target organ for diabetes-related damage. However, the lungs, and particularly the alveoli, have a complex capillary network that is subject to microvascular damage in diabetes. Diabetes affects the basal laminae of the alveoli and pulmonary capillaries, which in turn directly lowers oxygen uptake [1]. There are several possible mechanisms connecting Type 2 diabetes to lung function impairment including inflammation [2], reduced elastic recoil, hypoxia-induced insulin resistance and respiratory muscle neuropathy [3].

The relationship between Type 2 diabetes and respiratory conditions is established at clinical and epidemiological levels. High glycated haemoglobin (HbA1c) or poorly controlled diabetes is inversely related to Forced Vital Capacity (FVC) and Forced Expiratory Volume in one second (FEV1) in people with Type 2 diabetes [4–6]. High and uncontrolled HbA1c is more damaging to lung function than the duration of diabetes [5]. Moreover, restrictive lung functioning is predictive of an increased odds of Type 2 diabetes incidence [7]. Consistently, reduced FVC and FEV1, but not the FEV1/FVC ratio, are predictive of Type 2 diabetes development [8]. Restrictive lung conditions are associated with reduced FVC and a relatively normal or slightly reduced FEV1 as a percentage of their predicted values, leading to a normal FEV1/FVC Ratio (>70%). Obstructive conditions on the other hand are characterised by reduced FEV1 as a percentage of the predicted value (<80%), a relatively normal FVC percentage predicted, and consequently, a reduced FEV1/FVC ratio (<70%).

Despite the frequently reported association of Type 2 diabetes with airway restriction, the prevalence of obstructive lung conditions is high in those with diabetes. Chronic Obstructive Pulmonary Disease (COPD) is predicted to affect approximately 3 million people in the UK [9]. It is characterised by chronic airways obstruction, mainly in the form of a combination of emphysema and chronic bronchitis. It is the third highest cause of death globally [10] and typical symptoms include coughing and breathlessness. Approximately 10% of people with diabetes have COPD and hyperglycaemia may have a direct effect on COPD progression via lung inflammation and bacterial infections [11]. Asthma is another chronic disease of the lungs that is characterised by inflammation and narrowing of the airway and is commonly triggered by everyday allergens such as pollen. Symptoms include wheezing and tightness in the chest. There is a 2.2 hazard ratio of people with diabetes having asthma, compared to people without diabetes [12]. Inflammation and pro-inflammatory cytokines likely play a major role in the interaction between asthma and Type 2 diabetes [3].

The confounding factor in the development of obstructive lung disease in those with Type 2 diabetes is obesity. Recent studies have concluded that lower body mass index (BMI) is associated with COPD [13–15], with overweight and obesity acting as protective factors [14–17]. Yet, those with Type 2 diabetes, where overweight or obesity is commonplace, have significant risk of developing COPD. The protective factors of overweight and obesity in COPD are not upheld at BMI extremes [18]. Obesity has also been shown to negatively influence asthma outcomes by decreasing the efficiency of asthma treatment [19]. One meta-analysis investigating overweight, obesity and asthma reported that there is an increased odds of asthma development with increasing BMI in a dose-response manner [20].

Therefore, it remains unclear whether the association of asthma and COPD in people with Type 2 diabetes is being driven by hyperglycaemia or by obesity. This systematic review aims to assess the role of hyperglycaemia (Type 2 diabetes) and obesity on lung function in those with and without COPD or asthma.

## Methods

The Preferred Reporting Items for Systematic Reviews and Meta-Analysis (PRISMA) guidelines [21] were followed for this systematic review. The PRISMA checklists are available in Supporting Information S9 and S10 Files. The research question was formulated using the PICO (Population/Problem, Intervention/Exposure, Comparison, Outcome) framework for quantitative studies (Supporting Information, S1 File) [22].

### Search strategy

Two reviewers (R.L. and S.G.) methodically searched the Ovid MEDLINE and Embase online databases. The asthma and COPD searches were carried out separately using predefined keyword and Medical Subject Heading (MeSH) search terms relating to Type 2 diabetes, obesity, lung function and either asthma or COPD, and were modified accordingly for each database (Supporting Information, S2 File). The searches were restricted to articles written in English, journal articles, studies involving adults over the age 18, studies involving humans, and studies conducted from 2011 to 2024. Articles relating to SARS-CoV-2 were not included in this review.

A confirmed diagnosis of Type 2 diabetes was accepted as either HbA1c ≥ 48 mmol/mol, fasting plasma glucose ≥7.0 mmol/l, random plasma glucose ≥11.1 mmol/l or oral glucose tolerance test ≥11.1 mmol/l at 2 hours. COPD was diagnosed by an FEV1/FVC ratio of <70%. Asthma was diagnosed by an FEV1/FVC ratio of <70% with a post-bronchodilator improvement of 12% or more. Alternatively, a physician diagnosis was also accepted as confirmation. Self-reporting of disease was not accepted. Individuals were categorised according to BMI: lean BMI 18.5–24.9 kg/m$^2$, overweight BMI 25–29.9 kg/m$^2$ and obesity BMI ≥ 30 kg/m$^2$. A full list of acceptable paper diagnosis definitions of disease and BMI is available in Supporting Information, S3 File.

### Study selection and data extraction

Using Zotero v 6.0.9 (George Mason University, Virginia, USA), duplicate studies were removed, and an initial screen of titles and abstracts was performed (R.L. for COPD; S.G. and C.K. for asthma) based on inclusion and exclusion criteria. Inclusion criteria included: a confirmed diagnosis of Type 2 diabetes or asthma or COPD, and/or BMI classifications, and FEV1 and FVC spirometry data. Exclusion criteria were all other types of diabetes, obesity measured by anything other than BMI (i.e., waist circumference), any lung condition other than COPD or asthma, and case studies based on observations in a single individual. For the second screening step, the full manuscripts of all remaining studies were further assessed for eligibility (see Data Extraction, below) (R.L. for COPD; S.G., C.K. and R.L. for asthma). If the eligibility of any paper was questioned, the third reviewer (C.K.) was approached to discuss and resolve the query.

The Newcastle-Ottawa Scale (NOS) [23] was used to assess study quality for case-control and cohort studies. An adapted version of the scale [24] was used as a template for cross-sectional studies. This template was modified to suit

the needs of this review, available in Supporting Information, S4 File. Studies were deemed as low, medium, or high quality if they received 0–3, 4–5 or 6–7 stars, respectively, for cross-sectional studies. The case-control studies were deemed as low, medium, or high quality if they received 0–3, 4–6 or 7–9 stars, respectively. The cohort studies were deemed as low, medium, or high quality if they received 0–3, 4–7 or 8–10 stars, respectively. An additional star was given in the Outcome section for cohort studies as two stars were given when the spirometry values followed American Thoracic Society (ATS) and/or European Respiratory Society (ERS) guidelines, and one star was given when they did not state guideline criteria followed or when values were not properly recorded (available in Supporting Information, S5 File). Risk of Bias was used for interventional studies using the Cochrane risk-of-bias tool for randomised trials (RoB 2) methodology [25]. Studies were rated as having either low, some concerns or high risk of bias. All reviewers assessed the studies for quality and risk of bias (R.L. for COPD; S.G., C.K. and R.L. for asthma).

A pre-set, standardised data extraction form created by the reviewers was used to extract relevant information from each individual study. Data extracted included title, first author, year, study location, study type (case-control, cohort, cross-sectional, and interventional), the number of centres involved, participant numbers relevant to this review, age, sex, BMI, smoking status, FEV1% of predicted, FVC% of predicted and FEV1/FVC (L/L%) ratio. All statistical data obtained was taken in the format of mean ± standard deviation (SD).

## Statistical analysis

The mean ± SD of age, BMI, FEV1% of predicted, FVC% of predicted and FEV1/FVC (L/L%) were compiled from each study and the overall averages were calculated using the combined mean and combined standard deviations [26]. After data had been tested for normality using the Shapiro-Wilk test, the Kruskal-Wallis test with post hoc Dunn's multiple comparison test was used for non-parametric comparisons and Welch's ANOVA with Dunnett's T3 multiple comparison test for parametric comparisons between three or more groups. To compare two groups, Mann-Whitney test was used for non-parametric tests and t-test with Welch's correction was used for parametric tests. Multiple linear regression was conducted on never smokers without a lung disease for both the dependant variables FEV1% of predicted and FVC% of predicted using BMI, study region, age and Type 2 diabetes status as independent variables. The data was unable to be separated by sex and no other comparison of smoking status other than never smokers was available for regression analysis. Being from Europe or North America and not having Type 2 diabetes were used as the reference standards, as ethnicity data was insufficient for regression analysis. Latin America represents the West Indies and Brazil. Articles with missing data were excluded from relevant analysis if the necessary information was unavailable. Significance was accepted at $P < 0.05$. All statistical analysis was performed using GraphPad PRISM 10.1.2. (GraphPad Software, San Diego, California USA, www.graphpad.com).

## Results

### Selection of studies and study characteristics

After 15,104 papers were identified through the database search, 5,208 duplicates and 6 items (detected as ineligible by Zotero) were removed. The initial screen resulted in the removal of 9,463 items. The full text of 21 articles were not available after request. The remaining 406 articles were fully assessed for eligibility, with a final number of 93 suitable for inclusion in the review. Fig 1 shows the flow diagram of included and excluded articles based on the Preferred Reporting Items for Systematic Reviews and Meta-Analysis (PRISMA) guidelines. The full list of articles removed after the second screening is available in Supporting Information, S6 File.

Out of the 93 studies, 74 were cross-sectional studies, 13 were cohort studies, 4 were case-control studies and 2 were interventional. Over 30 study locations were recorded across Europe, North and South America, Asia, Australia, and Africa. India was the most frequently occurring study location at 30 counts. The studies involved 35,891 eligible participants, with 3,414 cases of Type 2 diabetes without any form of lung condition. Most articles consisted

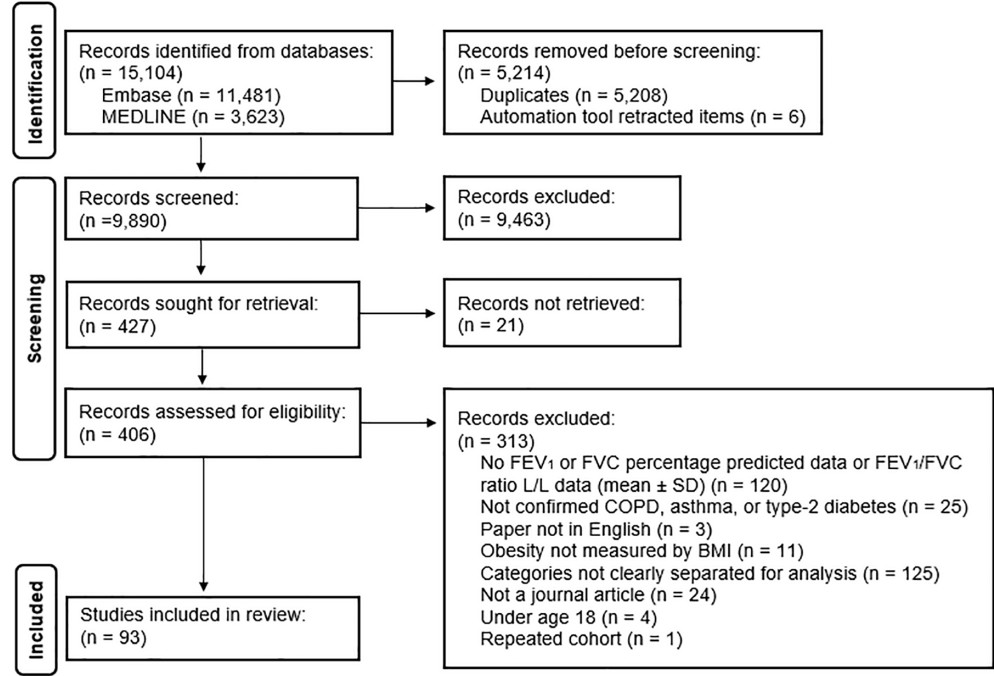

**Fig 1. PRISMA flow diagram for journal articles.** Shown are the included and excluded studies after literature search.

of both men and women, except for 11 studies that included only men and 6 that included only women. Information on all included studies is shown in Table 1, with all extracted data available in Supporting Information, S6 File. Quality assessment and risk of bias of studies revealed that all papers were of a medium or high quality for observational studies, and interventional studies had either low or some concerns for risk of bias. The summary of quality and risk of bias assessment scorings are available in Supporting Information, S7 File, with full information available in Supporting Information, S11 File.

### Association of Type 2 diabetes and/or obesity with lung function: Review of the literature

Initial analysis was performed on all studies, including 36 articles focused on Type 2 diabetes only (n = 3,414), 5 articles on Type 2 diabetes and COPD (n = 207), 1 article on Type 2 diabetes and asthma (n = 15), 23 obesity only articles (n = 3,795), 10 overweight only articles (n = 3,085), 22 lean weight articles (n = 3,704), 6 lean COPD weight articles (n = 6,600), 2 overweight COPD articles (n = 6,313), 8 obese COPD articles (n = 5,253), 12 lean asthma weight articles (n = 1,881), 8 overweight asthma articles (n = 1,049), and 11 obese asthma articles (n = 575). Due to the limited number of Type 2 diabetes studies containing a defined BMI range (n = 1 lean; n = 1 overweight; n = 3 obese) it was not possible to separate Type 2 diabetes data based on BMI and correlate it directly with lung function data, nor was it possible to perform meta-analysis. Therefore, all Type 2 diabetes data was grouped into a mixed BMI category and compared directly with non-Type 2 diabetes data from mixed BMI categories. For reference, approximately 64% of Type 2 diabetes study data had mean BMI values that fell into the overweight or obese categories, while approximately 61% of non-Type 2 study data fell within these ranges.

  Fifteen papers concluded that pulmonary decline in people with Type 2 diabetes followed a restrictive pattern [31,33,42,44,50,54,56–58,61,63,78,93,105,110]. However, one paper reported this association to be obstructive in nature

**Table 1. Study characteristics of review articles.**

| 1st Author, Year | Study Location | Study Type | No. of Centres | Participant Numbers (n) | Age (Mean±SD) | BMI (Mean±SD) | Sex (M/W n) |
|---|---|---|---|---|---|---|---|
| Katz et al. 2016 [27] | USA | Cohort | 1 | 580 COPD Ob (BMI ≥30) | 58.5±6.2 COPD Ob | – | 224/356 COPD Ob |
| Mishra et al. 2012 [28] | India | Cross Sectional | 1 | 15 T2D & COPD (Mixed BMI) / 15 T2D & Asthma (Mixed BMI) | 61.4±4.5 T2D & COPD / 60.3±7.5 T2D & Asthma | 21.7±4.0 T2D & COPD / 20.6±2.2 T2D & Asthma | 11/4 T2D & COPD / 7/8 T2D & Asthma |
| Wang et al. 2014 [29] | China | Cross Sectional | 1 | 37 T2D & COPD (Mixed BMI) / 215 T2D Only (Mixed BMI) | – | 27.1±5.2 T2D & COPD / 25.8±4.0 T2D Only | Both |
| Pandey et al. 2021 [30] | India | Cross Sectional | 1 | 43 Lean (BMI 18.5–22.9) | 26.0±6.1 Lean | 21.9±2.0 Lean | 43/0 Lean |
| Rani et al. 2019 [31] | India | Cross Sectional | 1 | 20 T2D Only (Mixed BMI) | 50.5±8.3 T2D Only | 25.9±3.1 T2D Only | Both |
| Shafiee et al. 2013 [32] | Iran | Cross Sectional | 1 | 80 T2D Only (Mixed BMI) | 53.6±11.9 T2D Only | 28.8±4.1 T2D Only | 55/25 T2D Only |
| S Ananthalakshmi et al. 2013 [33] | India | Cross Sectional | 1 | 30 T2D Only (Mixed BMI) | 44.8±8.9 T2D Only | 26.1±4.0 T2D Only | Both |
| Bhirange et al. 2020 [34] | India | Cross Sectional | 2 | 40 Lean (BMI 18.5–23) / 80 Ov (BMI 25–29.9) | 20.6±1.4 Lean / 20.8±1.5 Ov | 20.7±1.8 Lean / 26.8±1.3 Ov | 20/20 Lean / 40/40 Ov |
| Chidri and Ganji 2020 [35] | India | Cross Sectional | 1 | 100 T2D Only (Mixed BMI) | 43.5±6.2 T2D Only | 26.6±2.9 T2D Only | Both |
| Nowreen and Ahad 2019 [36] | India | Cross Sectional | 1 | 100 Ov (BMI 25–29.9) | 19.6±1.5 Ov | 26.89±2.6 Ov | 50/50 Ov |
| Attaur-Rasool and Shirwany 2012 [37] | Pakistan | Cross Sectional | 1 | 78 Lean (BMI 18.5–24.9) / 102 Ov (BMI 25–29.9) / 45 Ob (BMI ≥30) | 38.8±7.7 Lean / 41.3±8.0 Ov / 42.4±7.3 Ob | 22.3±1.9 Lean / 27.2±1.5 Ov / 32.4±2.6 Ob | Both |
| Yen et al. 2018 [38] | Malaysia | Cross Sectional | 1 | 40 Lean (BMI 18.5–25.9) / 40 Ob (BMI >30) | – | – | Both |
| Mittal et al. 2020 [39] | India | Cross Sectional | 1 | 101 T2D Only (Mixed BMI) | 51.5±8.4 T2D Only | 26.6±3.1 T2D Only | 52/49 T2D Only |
| Koraddi et al. 2015 [40] | India | Cross Sectional | 1 | 50 Lean (BMI 18.5–24.9) / 50 Ob (BMI >30) | 20.3±2.1 Lean / 20.7±2.3 Ob | 21.1±1.4 Lean / 30.4±1.1 Ob | 32/18 Lean / 23/27 Ob |
| Bhatti et al. 2019 [41] | Pakistan | Cross Sectional | 1 | 47 Lean (BMI 18.5–24.9) / 55 Ov (BMI 25–29.9) / 44 Ob (BMI ≥30) | – | – | Both |
| Saxena and Joshi 2020 [42] | India | Cross Sectional | 1 | 70 T2D Only (Mixed BMI) | 50.9±7.0 T2D Only | – | 37/33 T2D Only |
| Gutiérrez-Carrasquilla et al. 2019 [43] | Spain | Prospective Interventional | 1 | 60 T2D Only (Mixed BMI) | 58.1±6.4 T2D Only | 32.4±6.1 T2D Only | 47/13 T2D Only |
| Maan et al. 2021 [5] | Saudi Arabia | Case-Controlled Cross Sectional | 1 | 101 T2D Only (BMI <30) | 55.5±6.0 T2D Only | 25.0±2.1 T2D Only | 71/30 T2D Only |
| Naithok Jamatia et al. 2014 [44] | India | Cross Sectional | 1 | 30 T2D Only (Mixed BMI) | 57.7±4.7 T2D Only | 23.5±3.6 T2D Only | 19/11 T2D Only |
| Suresh Nayak et al. 2014 [45] | India | Cross Sectional | 1 | 40 Ob (BMI ≥30) | 49.5±6.9 Ob | 33.0±2.6 Ob | 40/0 Ob |

*(Continued)*

Table 1. (Continued)

| 1st Author, Year | Study Location | Study Type | No. of Centres | Participant Numbers (n) | Age (Mean±SD) | BMI (Mean±SD) | Sex (M/W n) |
|---|---|---|---|---|---|---|---|
| El-Shafey and El-Deib 2015 [46] | Egypt | Cross Sectional | 1 | 30 COPD Ob (BMI>30), 30 Asthma Ob (BMI>30) | 47.2±1.3 COPD Ob, 45.6±2.6 Asthma Ob | 34.6±2.2 COPD Ob, 36.6±3.0 Asthma Ob | 17/13 COPD Ob, 15/15 Asthma Ob |
| D'Ávila Melo et al. 2011 [47] | Brazil | Cross Sectional | 1 | 114 Ob (BMI≥30) | 37.0±11.3 Ob | – | 42/72 Ob |
| Bhattacharjee et al. 2018 [48] | Malaysia | Cross Sectional | 1 | 50 Lean (BMI 18.5–24.99), 50 Ob (BMI>30) | 21.2±2.2 Lean, 20.7±2.1 Ob | 22.8±2.3 Lean, 32.3±5.2 Ob | 0/50 Lean, 0/50 Ob |
| Khawaja 2011 [49] | Saudi Arabia | Cross Sectional | 1 | 68 Lean (BMI 18.5–24.9), 19 Ov (BMI 25–29.9), 26 Ob (BMI>30) | 19.8±0.8 Lean, 19.6±0.6 Ov, 20.0±0.7 Ob | – | 68/0 Lean, 19/0 Ov, 26/0 Ob |
| Shete and Garkal 2014 [50] | India | Cross Sectional Observational | 1 | 30 T2D Only (Mixed BMI) | 48.3±8.0 T2D Only | 27.7±4.1 T2D Only | Both |
| Röhling et al. 2018 [51] | Germany | Prospective Observational | 1 | 34 T2D Only (Mixed BMI) | 53±9 T2D Only | 30.8±5.6 T2D Only | 21/13 T2D Only |
| Zhang et al. 2021 [52] | China (USA Data) | Cross Sectional | Multiple | 2,198 Lean (BMI 18.5–24.9), 2,557 Ov (BMI 25–29.9), 2,483 Ob (BMI≥30) | 40.4 Lean, 44.9 Ov, 44.0 Ob | 22.42 Lean, 27.36 Ov, 35.26 Ob | 949/1,249 Lean, 1,470/1,087 Ov, 1,216/2,483 Ob |
| Bhattacharjee et al. 2018 [53] | Malaysia | Cross Sectional | 1 | 50 Lean (BMI 18.5–24.9), 50 Ob (BMI>30) | 21.2±2.2 Lean, 21.2±0.1 Ob | 22.3±2.4 Lean, 31.1±3.7 Ob | 50/0 Lean, 50/0 Ob |
| El-Azeem et al. 2013 [54] | Egypt | Cross Sectional | 14 | 30 T2D Only (BMI<30) | – | – | Both |
| Tesema et al. 2020 [55] | Ethiopia | Cross Sectional | 1 | 145 T2D Only (Mixed BMI) | 52.2±9.8 T2D Only | 26.5±3.1 T2D Only | 80/65 T2D Only |
| Shah et al. 2013 [56] | India | Cross Sectional | 1 | 60 T2D Only (Mixed BMI) | 53.9±8.5 T2D Only | – | 60/0 T2D Only |
| Bajaj et al. 2020 [57] | India | Case-Control | 1 | 100 T2D Only (Mixed BMI) | – | – | Both |
| Aparna 2013 [58] | India | Cross Sectional | 1 | 40 T2D Only (Mixed BMI) | 49.1±2.0 T2D Only | 25.2±1.6 T2D Only | 22/18 T2D Only |
| Kwon et al. 2012 [59] | Republic of Korea | Longitudinal Cohort | 1 | 207 T2D Only (Mixed BMI) | 42.6±5.6 T2D Only | 26.7±3.3 T2D Only | 207/0 T2D Only |
| Dharwadkar et al. 2011 [60] | India | Cross Sectional | 1 | 40 T2D Only (Mixed BMI) | 52.3±7.6 T2D Only | 22.7±3.4 T2D Only | 25/15 T2D Only |
| Zineldin et al. 2015 [61] | Egypt | Cross Sectional | 1 | 45 T2D Only (Mixed BMI) | 51.1±6.2 T2D Only | 24.5±0.8 T2D Only | 45/0 T2D Only |
| Pinto Pereira et al. 2013 [62] | West Indies | Cross Sectional | 2 | 109 T2D Only (Mixed BMI) | 55.6±11.3 T2D Only | 29.3±6.7 T2D Only | 47/62 T2D Only |
| Mandal et al. 2021 [63] | India | Cross Sectional | 1 | 100 T2D Only (BMI≤35) | 46.1±8.4 T2D Only | 25.6±2.9 T2D Only | 72/28 T2D Only |
| Al-Qerem et al. 2018 [64] | Jordan | Cross Sectional | 1 | 183 Lean (BMI 18.5–24.9), 202 Ob (BMI>30) | 30.1±11.7 Lean, 31.3±12.7 Ob | 22±1.9 Lean, 33.8±5.1 Ob | 106/77 Lean, 142/60 Ob |
| Wang et al. 2017 [65] | China | Cross Sectional | 1 | 457 Lean (BMI<24) | – | – | Both |
| Gabrielsen et al. 2011 [66] | Norway | Cross Sectional | 1 | 149 Ob (BMI≥35) | 43±11 Ob | 45.0±6.3 Ob | 35/114 Ob |

(Continued)

Table 1. (Continued)

| 1st Author, Year | Study Location | Study Type | No. of Centres | Participant Numbers (n) | Age (Mean±SD) | BMI (Mean±SD) | Sex (M/W n) |
|---|---|---|---|---|---|---|---|
| Vadasmiya and Patel 2019 [67] | India | Cross Sectional | 1 | 30 Lean (BMI <25) | 30.3±8.0 Lean | 22.6±1.8 Lean | 30/0 Lean |
| Klein et al. 2011 [68] | USA | Cross Sectional Retrospective | 1 | 76 T2D Only (Mixed BMI) | 63 T2D Only | 34.2 T2D Only | 33/43 T2D Only |
| Liu et al. 2021 [69] | China | Prospective Cohort | 1 | 55 COPD & T2D (Mixed BMI) | 70.5±6.7 COPD & T2D | 26.2±3.0 COPD & T2D | 55/0 COPD & T2D |
| Rodriguez et al. 2014 [70] | Spain | Cross Sectional | 9 | 108 COPD Ob (BMI≥30) | 68±8 COPD Ob | 33.2±2.8 COPD Ob | 102/6 COPD Ob |
| Mekov et al. 2016 [71] | Bulgaria | Cross Sectional | 1 | 53 COPD & T2D (Mixed BMI) | – | – | 40/13 COPD & T2D |
| Ora et al. 2011 [72] | Canada | Cross Sectional | 1 | 12 COPD Lean (BMI 18.5–24.9) / 12 COPD Ob (BMI 30.0–34.9) | 68±8 COPD Lean / 68±4 COPD Ob | 23.4±1.8 COPD Lean / 32.2±1.2 COPD Ob | 6/6 COPD Lean / 6/6 COPD Ob |
| Barbarito and De Mattia 2013 [73] | Italy | Cross Sectional | 1 | 16 COPD Lean (BMI 22–24.9) / 17 COPD Ob (BMI ≥40) | 73±8 COPD Lean / 69±7 COPD Ob | 23±1 COPD Lean / 47±6 COPD Ob | 10/6 COPD Lean / 7/10 COPD Ob |
| Figueira Gonçalves et al. 2020 [74] | Spain | Prospective Observational Cohort | 1 | 47 COPD & T2D (Mixed BMI) | 73.8±8.8 COPD & T2D | 30.3±4.2 COPD & T2D | 44/3 COPD & T2D |
| Galesanu et al. 2014 [75] | Canada | Cohort | 1 | 91 COPD Lean (BMI <25) | 65±9 COPD Lean | 22±3 COPD Lean | 76/15 COPD Lean |
| Wytrychowski and Hans-Wytrychowska 2016 [76] | Poland | Cross Sectional | 1 | 29 COPD Ob (BMI >30) / 28 Asthma Ob (BMI >30) | 68.3±7.8 COPD Ob / 52.3±12.8 Asthma Ob | 33.4±3.1 COPD Ob / 33.6±3.4 Asthma Ob | 16/13 COPD Ob / 10/18 Asthma Ob |
| O'Donnell et al. 2011 [77] | Canada | Cross Sectional Retrospective | 1 | 733 COPD Lean (BMI 18.5–24.9) / 804 COPD Ov (BMI 25–29.9) / 654 COPD Ob (BMI ≥30) | 65.8±9.4 COPD Lean / 65.5±9.4 COPD Ov / 63.4±9.5 COPD Ob | 22.4±1.7 COPD Lean / 27.4±1 COPD Ov / 34.8±4.6 COPD Ob | 387/346 COPD Lean / 549/255 COPD Ov / 408/246 COPD Ob |
| Karintholil et al. 2021 [78] | India | Cross Sectional | 1 | 80 T2D Only (Mixed BMI) | 58.4±10.7 T2D Only | 24.7±3.5 T2D Only | 45/35 T2D Only |
| Fuso et al. 2015 [79] | Italy | Longitudinal Cohort | 1 | 45 T2D Only (Mixed BMI) | 63.8±6.4 T2D Only | 29.5±5.0 T2D Only | 28/17 T2D Only |
| Huang et al. 2014 [80] | China | Cross Sectional Retrospective | 1 | 292 T2D Only (Mixed BMI) | 66.9±9.6 T2D Only | 23.9±3.8 T2D Only | 181/111 T2D Only |
| Zakaria et al. 2019 [81] | Malaysia | Cross Sectional Retrospective | 1 | 53 Lean (BMI 18.5–24.9) / 43 Ov (BMI 25–29.9) / 17 Ob (BMI ≥30) | 61.0±14.2 Lean / 56.1±12.3 Ov / 56.8±11.2 Ob | – | 36/17 Lean / 30/13 Ov / 9/8 Ob |
| Putcha et al. 2022 [82] | USA (TIOSPIR 9 countries) (UPLIFT 37 countries) | Post hoc analysis (Cross Sectional) | 117 TIOSPIR 490 UPLIFT | 5,721 COPD Lean (BMI 20–24.9) / 5,509 COPD Ov (BMI 25–29.9) / 3,823 COPD Ob (BMI ≥30) | 65.5±9.2 COPD Lean / 65.4±9.0 COPD Ov / 63.9±8.7 COPD Ob | – | 4,169/1,552 COPD Lean / 4,037/1,472 COPD Ov / 2,582/1,241 COPD Ob |

(Continued)

Table 1. (Continued)

| 1st Author, Year | Study Location | Study Type | No. of Centres | Participant Numbers (n) | Age (Mean±SD) | BMI (Mean±SD) | Sex (M/W n) |
|---|---|---|---|---|---|---|---|
| Kwon et al. 2012 [83] | Republic of Korea | Cohort | 11 | 546 Asthma Lean (BMI 18.5–24.9); 233 Asthma Ov (BMI 25–29.9); 42 Asthma Ob (BMI ≥30) | 45.5±15.5 Asthma Lean; 49.5±15.9 Asthma Ov; 53.4±14.3 Asthma Ob | — | 238/308 Asthma Lean; 126/107 Asthma Ov; 11/31 Asthma Ob |
| Barranco et al. 2011 [84] | Spain | Cross Sectional | 1 | 151 Asthma Lean (BMI <25); 72 Asthma Ov (BMI 25–29.9); 28 Asthma Ob (BMI ≥30) | — | — | 34/117 Asthma Lean; 37/35 Asthma Ov; 12/16 Asthma Ob |
| Tu et al. 2015 [85] | China | Cross Sectional Retrospective | 1 | 32 T2D Only (Mixed BMI) | 45.1±11.3 T2D Only | 30.7±3.5 T2D Only | 14/18 T2D Only |
| Raviv et al. 2011 [86] | USA | Cross Sectional | 38 | 56 Asthma Lean (BMI 20–24.9); 69 Asthma Ov (BMI 25–29.9); 101 Asthma Ob (BMI ≥30) | 35.3±13.6 Asthma Lean; 37.0±13.6 Asthma Ov; 38.6±10.9 Asthma Ob | 22.6±1.4 Asthma Lean; 27.3±1.6 Asthma Ov; 37.3±6.4 Asthma Ob | 18/38 Asthma Lean; 26/43 Asthma Ov; 23/78 Asthma Ob |
| Melo et al. 2016 [87] | Brazil | Prospective Longitudinal Cohort | 1 | 43 Ob (BMI ≥35) | 38.7±10.1 Ob | 44.2±7.5 Ob | 13/30 Ob |
| Razi et al. 2014 [88] | Iran | Cross Sectional | 1 | 107 Asthma Lean (BMI <25) | 39.5±14.3 Asthma Lean | 22.1±2.2 Asthma Lean | 71/36 Asthma Lean |
| Peixoto-Souza et al. 2013 [89] | Brazil | Cross Sectional | 1 | 37 Lean (BMI 18.5–24.9); 72 Ob (BMI 40–55) | 34.9±7.6 Lean; 34.6±6.8 Ob | 22.7±1.9 Lean; 45.8±5.4 Ob | 0/37 Lean; 0/72 Ob |
| Steier et al. 2014 [90] | UK | Cross Sectional | 1 | 9 Lean (BMI 18.5–24.9); 9 Ob (BMI >30) | 38±11 Lean; 45±13 Ob | 23.2±1.6 Lean; 46.8±17.2 Ob | 5/4 Lean; 4/5 Ob |
| Pisi et al. 2012 [91] | Italy | Cohort | 1 | 203 Asthma Lean (BMI 18.5–24.9); 145 Asthma Ov (BMI 25–30) | 37±15 Asthma Lean; 48±16 Asthma Ov | — | Both |
| Al-Khlaiwi et al. 2021 [92] | Saudi Arabia | Cross Sectional | 2 | 110 T2D Only (Mixed BMI) | 45.5±13.0 T2D Only | 28.6±4.9 T2D Only | 71/39 T2D Only |
| He et al. 2020 [93] | China | Cross Sectional | 1 | 326 T2D Only (Mixed BMI) | 53.5±11.6 T2D Only | 27.1±3.9 T2D Only | 222/104 T2D Only |
| López-Cano et al. 2017 [94] | Spain | Case-Control | 1 | 98 Ob (BMI ≥30); 49 T2D Ob (BMI ≥30) | 48.5±9.4 Ob; 51.3±10.6 T2D Ob | 42.6±6.7 Ob; 42.0±7.7 T2D Ob | 24/74 Ob; 12/37 T2D Ob |
| Agondi et al. 2012 [95] | Brazil | Cross Sectional | 1 | 153 Asthma Lean (BMI 18.5–24.9); 153 Asthma Ov (BMI 25–29.9); 145 Asthma Ob (BMI ≥30) | 45.8±17.8 Asthma Lean; 50.4±14.6 Asthma Ov; 51.6±13.6 Asthma Ob | 22.7±1.7 Asthma Lean; 27.3±1.3 Asthma Ov; 34.0±3.7 Asthma Ob | 45/108 Asthma Lean; 35/118 Asthma Ov; 22/123 Asthma Ob |
| Sonpeayung et al. 2019 [96] | Thailand | Cross Sectional | 1 | 20 Lean (BMI 18.5–22.9) | 27.2±3.9 Lean | 21.6±1.0 Lean | 20/0 Lean |

(Continued)

Table 1. (Continued)

| 1st Author, Year | Study Location | Study Type | No. of Centres | Participant Numbers (n) | Age (Mean±SD) | BMI (Mean±SD) | Sex (M/W n) |
|---|---|---|---|---|---|---|---|
| Shimray et al. 2014 [97] | India | Cross Sectional | 2 | 27 COPD Lean (BMI ≥18.5) | – | – | Both |
| Wei et al. 2011 [98] | Taiwan | Cohort | 1 | 94 Ob (BMI >32) | 31.2±9.8 Ob | 43.4±7.3 Ob | Both |
| Farah et al. 2011 [99] | Australia | Cohort | 1 | 20 Asthma Lean (BMI 18.5–24.9) 14 Asthma Ov (BMI 25–29.9) 15 Asthma Ob (BMI ≥30) | 29 Asthma Lean 44 Asthma Ov 39 Asthma Ob | – | 7/13 Asthma Lean 10/4 Asthma Ov 9/6 Asthma Ob |
| Rasslan et al. 2015 [100] | Brazil | Cross Sectional | 1 | 25 Lean (BMI 18.5–24.9) 28 Ov (BMI 25–29.9) 27 Ob (BMI ≥30) | 31.2±7.1 Lean 31.9±6.9 Ov 32.9±7.3 Ob | 22.4±1.6 Lean 27.8±1.3 Ov 33.3±2.2 Ob | 0/25 Lean 0/28 Ov 0/27 Ob |
| Shenoy et al. 2011 [101] | India | Cross Sectional | 1 | 48 Lean (BMI 18–25) | 20.8±2.6 Lean | 21.6±1.5 Lean | 48/0 Lean |
| Divya et al. 2022 [102] | India | Cross sectional | 1 | 50 Lean (BMI 18–22.9) | – | 19.8±1.4 Lean | 0/50 Lean |
| Jorres et al. 2022 [103] | Germany | Cross sectional | 1 | 23 Lean (BMI <25) 66 Ob (BMI ≥30) | 36.3±14.4 Lean 49.9±13.2 Ob | 21.4±1.9 Lean 37.7±7.0 Ob | 3/20 Lean 18/48 Ob |
| Kumar et al. 2022 [104] | India | Cross sectional | 1 | 61 Ov (BMI 25–29.9) 17 Ob (BMI ≥30) | – | – | Both |
| Vasantha et al. 2022 [105] | India | Case Control | 1 | 25 T2D Only (BMI <30) | – | – | 15/10 T2D Only |
| Pekince and Baccioglu 2022 [106] | Turkey | Case Control | 1 | 22 Asthma Lean (BMI <24) | – | – | Both |
| Vennilavan et al. 2022 [107] | India | Prospective observational study | 1 | 90 Asthma Lean (BMI <25) | 39.63±14.38 Asthma Lean | 21.21±2.61 Asthma Lean | Both |
| Bermúdez Barón et al. 2022 [108] | Sweden | Cohort | 5 | 485 Asthma Lean (BMI 18.5–25) 327 Asthma Ov (BMI 25–30) 133 Asthma Ob (BMI ≥30) | 37.7±11.4 Asthma Lean 43.3±11.2 Asthma Ov 43.3±11.2 Asthma Ob | 22.6±1.7 Asthma Lean 27.2±1.4 Asthma Ov 33.0±3.0 Asthma Ob | 182/303 Asthma Lean 182/145 Asthma Ov 55/78 Asthma Ob |
| Yadav et al. 2022 [109] | India | Cross sectional | 2 | 216 T2D Only (Mixed BMI) | – | – | 141/75 T2D Only |
| Abdulsaied et al. 2022 [110] | Iraq | Cross sectional | 1 | 160 T2D Only (BMI <40) | 54.6±7.9 T2D Only | 28.7±4.6 T2D Only | 84/76 T2D Only |
| Cao et al. 2022 [111] | Republic of Korea | Cross sectional | 1 | 83 Asthma Lean (BMI 18.5–24.9) 36 Asthma Ov (BMI 25–29.9) 10 Asthma Ob (BMI ≥30) | 49.4±14.5 Asthma Lean 54.5±14.4 Asthma Ov 40.9±14.6 Asthma Ob | 21.9±1.7 Asthma Lean 26.5±1.2 Asthma Ov 32.6±4.4 Asthma Ob | 24/59 Asthma Lean 12/24 Asthma Ov 7/3 Asthma Ob |
| Gurudatta Pawar et al. 2022 [112] | India | Cross sectional | 1 | 80 T2D Only (Mixed BMI) | 51.1±10.2 T2D Only | 25.1±3.6 T2D Only | 43/37 T2D Only |

(Continued)

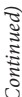

Table 1. (Continued)

| 1st Author, Year | Study Location | Study Type | No. of Centres | Participant Numbers (n) | Age (Mean ± SD) | BMI (Mean ± SD) | Sex (M/W n) |
|---|---|---|---|---|---|---|---|
| Mankar et al. 2022 [113] | India | Cross sectional | 1 | 105 Lean (BMI 18–24.9) 40 Ov (BMI 25–29.9) 22 Ob (BMI ≥30) | – | – | Both |
| Perossi et al. 2022 [114] | Brazil | Cross sectional | 1 | 37 Ob (BMI ≥40) | – | – | 0/37 Ob |
| Lopez-Cano et al. 2022 [115] | Spain | Randomised Controlled Trial | 5 | 76 T2D Ob (BMI ≥30) | 58.6 ± 7.5 T2D Ob | 34.8 ± 4.1 T2D Ob | 46/30 T2D Ob |
| Bourdin et al. 2023 [116] | France | Cohort | 1 | 13 Asthma Ob (BMI ≥30) | – | 36.3 ± 5.8 Asthma Ob | 0/13 Asthma Ob |
| Ricketts et al. 2023 [117] | UK | Cross sectional | Multiple | 25 Asthma Lean (BMI <25) | – | – | 8/17 Asthma Lean |
| Ashrith et al. 2022 [118] | India | Cross sectional | 1 | 100 T2D Only (Mixed BMI) | 59.9 ± 9.7 T2D Only | 24.9 ± 3.6 T2D Only | Both |

Study characteristics of all 93 articles included in the review. Participant numbers show only numbers that were suitable for this review, classified into body mass index (BMI) categories. Age and BMI are presented as mean ± standard deviation. *T2D* Type 2 Diabetes; *COPD* Chronic Obstructive Pulmonary Disease; *SD* Standard Deviation; *M/W* Men/Women; *Ov* overweight; *Ob* obese. Lean BMI 18.5–24.9 kg/m², overweight BMI 25–29.9 kg/m², obese BMI ≥30 kg/m².

[80] and another concluded that newly diagnosed Type 2 diabetes is also more representative of an obstructive pattern [51]. Two papers proposed various biomarkers for pulmonary function in people with T2DM without a lung disease: namely serum SP-D concentrations in people with Type 2 diabetes and obesity as a lung function biomarker [94], and postural position measuring diffusing capacity of the lungs for carbon monoxide (DLCO) as a test for early microvascular damage in people with T2DM [112]. Eight papers reported a negative association of FVC and FEV1 with diabetes duration [31,39,42,54,60,61,78,105]. while three reported no association with duration of disease [56,57,118]. Eight papers reported lower lung function in people with Type 2 diabetes compared to people without [32,35,51,55,59,68,109,115]. Ten articles concluded that uncontrolled glycaemic states can directly exacerbate lung function decline [5,43,44,51,60–62,79,92,105]. Opposingly, two studies reported no association of lung function in people with T2DM with glucose control [35,56]. One article suggested the restrictive pattern seen within people with Type 2 diabetes is correlated with insulin resistance [57] and finally, one article reported that women have both lower FEV1 and FVC in comparison to men with Type 2 diabetes [92].

In articles covering Type 2 diabetes and a lung disease, two articles suggested that uncontrolled glucose results in a worse outcome for people with Type 2 diabetes and COPD [29,71], while Liu et al. (2021) [69] states that there is worse lung performance in people with COPD in the presence of Type 2 diabetes than just COPD alone. Figueira Gonçalves et al. (2020) [74] concluded that in people with Type 2 diabetes and COPD, the risk of hospitalisation is greater. Finally, Mishra et al. (2012) [28] concluded that in people with Type 2 diabetes and diagnosed asthma or COPD, there is a restrictive rather than obstructive pattern of lung impairment, which is unexpected considering they are both considered obstructive lung diseases. Additionally, this study also found a negative association of FEV1 and FVC with the duration of diabetes in people with Type 2 diabetes and asthma.

In obese people without Type 2 diabetes, lung function is reduced compared to lean controls [34,37,45,52,64,90,102]. Five papers reported that lung function decline in obese people presents in a restrictive pattern [30,41,67,81,101]. However, two articles state that this association is obstructive [38,103]. Two papers concluded that BMI is negatively correlated with pulmonary function tests [49,113], and another stated this was only true for FVC [65]. Four articles stated there is no correlation between BMI and FEV1 and FVC [30,36,48,53] and one study concluded that lung function parameters are only influenced by BMI when BMI is ≥ 45 kg/m² [47]. In fact, six articles have stated that central obesity is a better determinant of lung function decline than BMI [34,48,53,66,85,96]. Opposingly, one study determined that subcutaneous fat was more detrimental to lung function than central adiposity in obese women [100]. In terms of gender differences, obese women have significantly lower lung function than obese men [36,40,81,113]. Melo et al. (2016) [87] concluded that morbid obesity causes accelerated lung aging, but that weight loss can reverse this. One other paper supports the conclusion that weight loss in obese individuals can improve lung functions and volumes [98]. Lastly, in the final two obesity only studies, Kumar et al. (2022) [104] concluded that FEV1 and FVC spirometry values were higher in rural communities and lower in urban areas, and Perossi et al. (2022) showed that impulse oscillometry is a better technique for detecting airway resistance and obstruction than spirometry [114].

Overall, overweight and obesity in people with asthma results in reduced FEV1 and FVC compared to lean controls [91,106,108]. Two papers state that asthma severity is not impacted by obesity [84,86]. However, oppositely, Farah et al. (2011) [99] showed that BMI independently increases the risk of poor asthma control, Agondi et al. (2012) [95] determined a negative association between FEV1 and BMI in obese people with asthma, Bourdin et al. (2023) [116] concluded that small-airway responsiveness in obese women with asthma is negatively affected by BMI and Kwon et al. (2012) showed that there is increased wheezing in obese people with asthma [83]. Additionally, people with asthma who are obese have poorer response to treatment compared to those who are not obese [88]. Lung age is significantly higher in women with morbid obesity [89], while the duration of asthma was more influential for lung function decline than BMI in elderly people over 60 years of age [95]. Weight loss and exercise in obese people with asthma can significantly improve FEV1 and FVC values [46,117]. Two papers discussed potential biomarkers of lung function decline in obese people with asthma.

Vennilavan *et al.* (2022) [107] reported that FEV1 was negatively associated with the pro-inflammatory markers IL-8 and positively with IL-5 and Cao *et al.* (2022) [111] reported low levels of serum surfactant protein B in obese people with asthma. One study reported that obese women with asthma have worse asthma control than men [76].

Obesity in people with COPD is associated with worse lung function [27,72]. However, one study states that lung function is only associated with obesity in GOLD stages 3–4 [77]. Furthermore, weight loss in obese people with COPD improves airway obstruction [46]. However, four studies have concluded that obesity does not negatively affect lung function in people with COPD, in fact, it can even improve it [70,73,82,97]. Other papers contradict this and say that the protective nature of obesity in people with COPD is not upheld at extreme BMIs of >35 kg/m$^2$ [18] and that protection against mortality is due to muscle mass, exercise capacity and preserved lung function, as opposed to fat accumulation [75]. Finally, Wytrychowski et al. (2016) reported age to be a risk factor for obesity in people with COPD [76].

### Airway restriction in people with Type 2 diabetes

Age was significantly different for each COPD BMI category versus their respective controls (Fig 2). FVC did not differ between BMI categories (Supporting Information, S8 File). People with Type 2 diabetes who did not have a co-morbid lung condition had apparent normal lung function (80.46 ± 13.06 FEV1% of predicted; 80.1 ± 13.81 FVC% of predicted; 86.26 ± 6.17 FEV1/FVC (L/L%); Fig 3). However, they had a reduction in FEV1% of predicted of 11.9% ($P < 0.0001$) and FVC% of predicted of 12.2% ($P < 0.01$) compared to those without Type 2 diabetes (Fig 3). FEV1/FVC ratio was not significant. This would suggest that lung function in people with T2DM without a lung disease is restrictive in nature.

In those with COPD, the FVC% of predicted was significantly lower in people with Type 2 diabetes compared to those without ($P < 0.05$; Fig 3). Those with COPD had a characteristically normal obstructive phenotype (reduced FEV1 (<80%), relatively normal FVC (~80%), and reduced FEV1/FVC (<0.7)). Individuals with asthma mostly presented with normal lung function (81.19 ± 10.55 FEV1% of predicted; 89.95 ± 9.19 FVC% of predicted). However, those with asthma and Type 2 diabetes had lower lung function than those with asthma alone (mean FEV1% of predicted 63.07; mean FVC% of predicted 73.07). However, interpretation of this finding is limited as only one study was available (n = 1; Fig 3). Obesity in those without Type 2 diabetes did not have an impact on lung function or age data (Fig 4).

In the multiple linear regression analysis on never smokers, having a Type 2 diabetes diagnosis and being in the geographical region of Asia were both significantly associated with reduced FEV1% of predicted (Table 2) and FVC% of predicted (Table 3). Multiple combinations of the model were run, with Type 2 diabetes associated with 23.51% ($P$ 0.0024, 95% CI −37.61 to −9.42) and 23.67% ($P$ 0.0007, 95% CI −35.57 to −11.77) lower FEV1% of predicted and FVC% of predicted, respectively, compared to not having Type 2 diabetes in the best fit model. Being from the region of Asia was associated with 16.57% ($P$ 0.0409, 95% CI −32.39 to −0.76) and 21.36% ($P$ 0.0213, 95% CI −39.09 to −3.62) lower FEV1% of predicted and FVC% of predicted, respectively, compared to European/North American individuals in the best fit model. BMI was not a significant contributor to lung function in any of the models. FEV1/FVC models exhibited a poor fit for multiple linear regression (Supporting Information, S12 File). All data was normally distributed, except for model 2 in the FVC regression. The variance inflation factor (VIF) is presented for examination of multicollinearity.

### Discussion

In this systematic review, the impact of Type 2 diabetes and obesity on lung function in those with and without COPD or asthma was investigated. A role for Type 2 diabetes in reducing lung function independent of a diagnosis of asthma or COPD has been shown. Although the decline in lung function in Type 2 diabetes did not meet the clinical diagnosis of restrictive disease, the pattern was consistent with airways restriction. Once asthma or COPD was established, neither hyperglycaemia nor increasing BMI further exacerbated lung decline significantly, irrespective of Type 2 diabetes diagnosis. A diagnosis of Type 2 diabetes and studies conducted in Asia were both significantly linked to reductions in FEV1% of predicted and FVC% of predicted, while BMI had no significant impact.

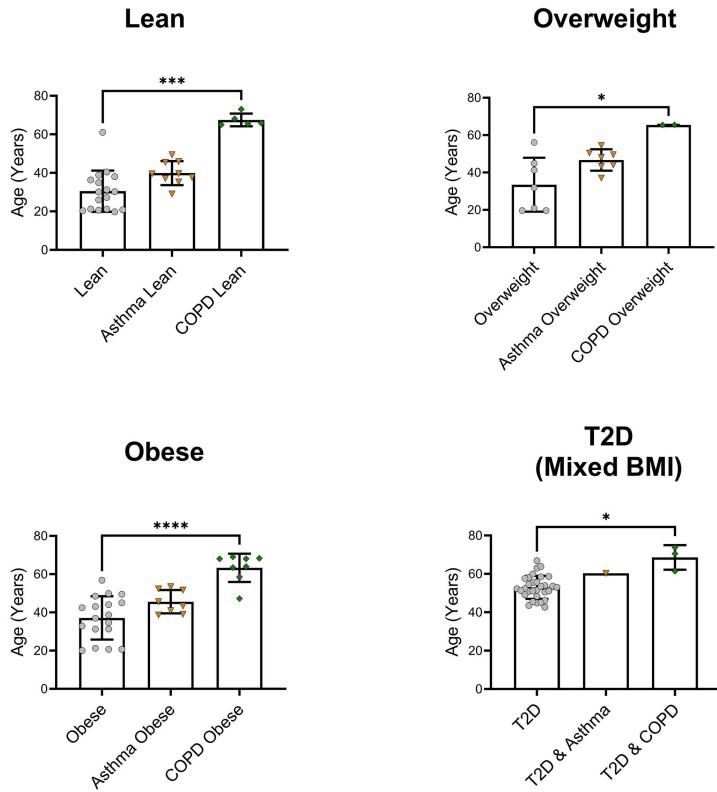

**Fig 2. Age data separated by body mass index (BMI) status.** Data are presented as mean ± standard deviation. *COPD* chronic obstructive pulmonary disease; *T2D* Type 2 diabetes. * $P \le 0.05$, *** $P \le 0.001$, **** $P \le 0.0001$. Lean BMI 18.5-24.9 kg/m², overweight BMI 25-29.9 kg/m², obese BMI ≥ 30 kg/m².

A prior meta-analysis has shown a mild restrictive phenotype in people with Type 2 diabetes [119]. However, it only included participants without a respiratory disease. This analysis shows that having a lung disease concurrently with Type 2 diabetes does not exacerbate lung impairment in comparison to those who have a lung disease in the absence of diabetes. Therefore, it is unlikely that outcomes for people with Type 2 diabetes with asthma or COPD, compared to those with asthma or COPD alone, will differ significantly. However, this finding may be influenced by the use of corticosteroids for the treatment of both asthma and COPD. Corticosteroids are anti-inflammatory drugs that exacerbate glucose control in those with preexisting diabetes (steroid-induced hyperglycaemia) [120] and in those without preexisting diabetes (steroid-induced diabetes) [121]. In people with COPD and asthma, there is a dose-dependent increase in HbA1c levels associated with inhaled corticosteroid therapy [122–124]. Furthermore, inhaled corticosteroid use is associated with an increased rate of diabetes and diabetes progression [125]. Despite the potential impact of steroid use on diabetes outcomes, lung function will likely improve. Therefore, lung function data between those with and without Type 2 diabetes with a co-morbid lung disease may balance. Interestingly, FVC% of predicted in people with Type 2 diabetes and COPD was significantly reduced compared to those with COPD alone. As corticosteroid use should in this instance be equal, FVC in people with Type 2 diabetes is still markedly reduced even with lung function interventions compared to those with COPD only. In those without a diagnosed lung disease, where corticosteroid use is not a confounding factor, both FEV1% of predicted and FVC% of predicted were significantly reduced in those with Type 2 diabetes compared to people without Type 2 diabetes.

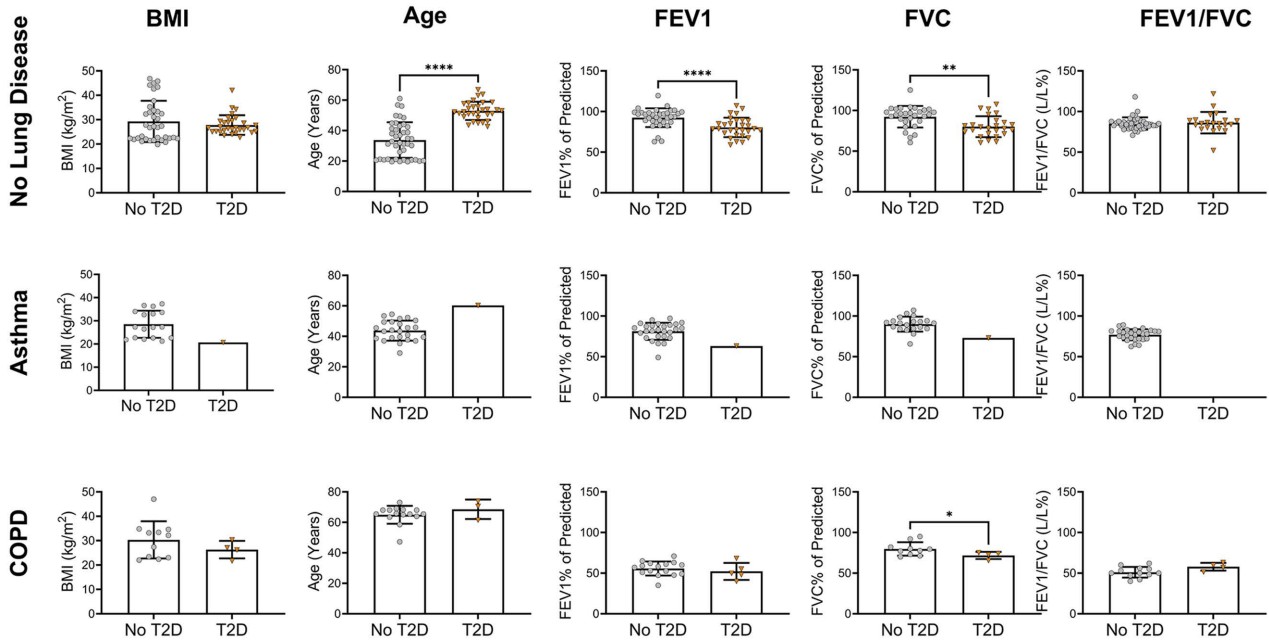

**Fig 3. Comparison of those with Type 2 diabetes to those without according to lung disease.** Figure shows statistical significance for age, body mass index (BMI) forced expiratory volume in one second (FEV1; % of predicted), forced vital capacity (FVC; % of predicted) and FEV1/FVC (L/L%) ratio separated by lung disease status. Data is not separated by BMI. Data are presented as mean ± standard deviation. *COPD* chronic obstructive pulmonary disease. * $P \le 0.05$, ** $P \le 0.01$, **** $P \le 0.0001$. Lean BMI 18.5-24.9 kg/m², overweight BMI 25-29.9 kg/m², obese BMI ≥ 30 kg/m².

Having a diagnosis of Type 2 diabetes and being from the geographical region of Asia were independent risk factors for exacerbation of lung function in this analysis. Obesity in the presence of COPD has a clinically relevant obstructive phenotype in this analysis, but this did not differ from individuals who had COPD and were lean. This is somewhat in contrast to the literature, which has inconsistencies in the protective nature of obesity in COPD known as 'the obesity paradox'. For example, Ora *et al.* (2009) [126] concluded that there is no significant difference in dyspnoea between obese people with COPD and lean COPD individuals, while opposingly, Cecere *et al.* (2011) [127] concluded that dyspnoea in obese COPD people is nearly 5 times worse than in lean individuals (adjusted OR 4.91; $P = 0.002$). A recent meta-analysis investigating obesity in COPD has reported that no conclusion of the role of obesity in COPD can be elucidated with the current conflicting literature [128].

In this analysis, BMI had no significant impact on lung function in people without a preexisting lung condition. This is generally in contrast to existing literature [34,37,45,52,64,90,102]. The findings in this article may be due to the vastly different population ethnicities in the included studies. Additionally, central obesity may be a stronger indicator of lung function than BMI [53]. Despite this finding, weight-loss improves spirometry measurements in obese people both with and without asthma [129,130]. In the presence of a lung disease, BMI not having a significant impact on lung function may be explained by the fact that lung function has already been diminished as a result of the underlying lung disease. Significant impairment is already occurring and therefore limiting the capacity for further deterioration. Consequently, the underlying lung disease pathology may be eclipsing the impact that obesity, and indeed T2DM, may be having on the lungs. In the presence of Type 2 diabetes, as the data in this instance could not be analysed according to BMI, the linear regression analysis was performed to address this.

Pathophysiological mechanisms of pulmonary impairment in Type 2 diabetes are not fully elucidated. The complex mechanisms driving pulmonary dysfunction in people with Type 2 diabetes are thought to include basal lamina thickening

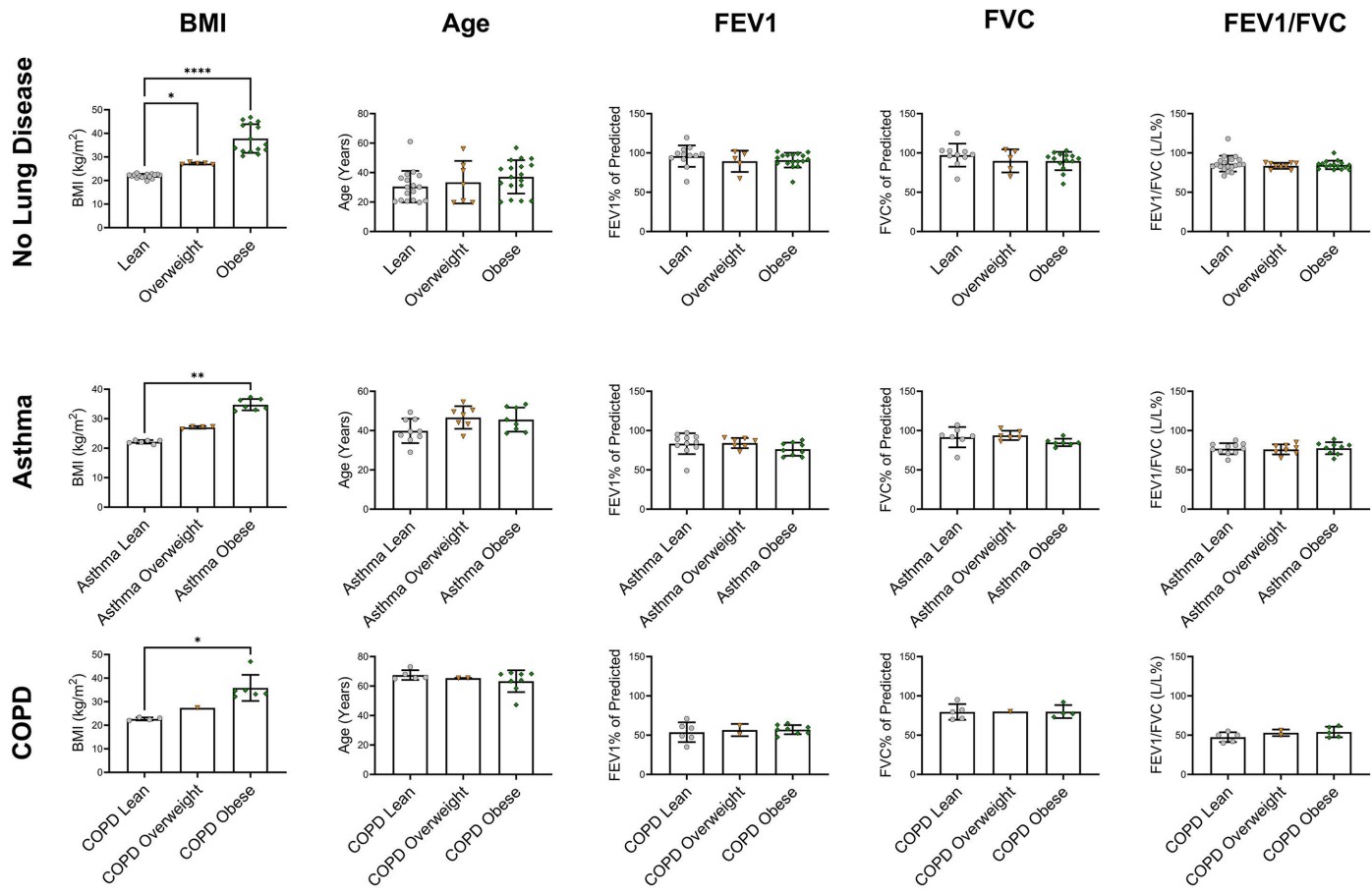

**Fig 4. Comparison between those without Type 2 diabetes according to BMI and lung disease.** Figure shows statistical significance diabetes for age, body mass index (BMI), forced expiratory volume in one second (FEV1; % of predicted), forced vital capacity (FVC; % of predicted) and FEV1/FVC (L/L%) ratio separated by lung disease status. Data are presented as mean±standard deviation. *COPD* chronic obstructive pulmonary disease. * $P \leq 0.05$, ** $P \leq 0.01$, **** $P \leq 0.0001$. Lean BMI 18.5-24.9 kg/m$^2$, overweight BMI 25-29.9 kg/m$^2$, obese BMI ≥ 30 kg/m$^2$.

[131], respiratory muscle dysfunction [132], diabetic neuropathy [133], advanced glycation of end-products [134,135], systemic inflammation [136], microangiopathy [137] and surfactant protein dysregulation [138]. It has been reported that lung decline, especially forced vital capacity, is accelerated after a diabetes diagnosis in those without a pre-existing lung condition [139]. For that reason, lung monitoring in Type 2 diabetes may be warranted to prevent any deterioration of lung function.

There are a number of limitations to this systematic review. Most of the included studies were cross-sectional studies. The data for these studies were taken at only one time point and cannot account for any causal relationship. Thus, although many cross-sectional studies conclude there is accelerated lung dysfunction in diabetes, some longitudinal studies suggest there is no acceleration in decline after diabetes diagnosis [140]. Heterogeneity in study methodologies can significantly impact the consistency of outcomes across included studies, as is seen here in the recording of FEV1/FVC measurements. Typically, restrictive and obstructive lung diseases are diagnosed by FEV1/FVC (L/L) ratio, and not by FEV1/FVC% of predicted. Unfortunately, not all studies stated which measurement they used; therefore, if the measurement used was not clear, it was assumed to be L/L. The Type 2 diabetes and asthma group had insufficient study

**Table 2. Multiple linear regression of FEV1% of predicted.**

| Model | Predictor | Estimate (β) | SE | 95% CI | VIF | P value | Goodness of Fit (R²) |
|---|---|---|---|---|---|---|---|
| 1 | Intercept | 93.41 | 2.13 | [89.09, 97.72] | | <0.0001 | |
| | T2DM [Yes] | −14.32 | 3.05 | [-20.50, -8.14] | 1.000 | <0.0001 | 0.3733 |
| 2 | Intercept | 90.40 | 9.71 | [70.39, 110.4] | | <0.0001 | |
| | BMI | 0.12 | 0.31 | [-0.51, 0.76] | 1.046 | 0.6938 | |
| | T2DM [Yes] | −14.93 | 4.12 | [-23.41, -6.44] | 1.046 | 0.0013 | 0.3677 |
| 3 | Intercept | 120.10 | 14.56 | [89.84, 150.4] | | <0.0001 | |
| | Region [Asia] | −18.31 | 7.14 | [-33.17, -3.46] | 3.799 | 0.0181 | |
| | Region [Middle East] | −2.42 | 8.48 | [-20.05, 15.22] | 2.148 | 0.7784 | |
| | Region [Africa] | −14.75 | 11.55 | [-38.77, 9.27] | 1.434 | 0.2155 | |
| | Region [Latin America] | −13.57 | 8.23 | [-30.69, 3.55] | 2.590 | 0.1143 | |
| | BMI | −0.34 | 0.33 | [-1.03, 0.34] | 1.511 | 0.3072 | |
| | T2DM [Yes] | −19.86 | 4.41 | [-29.02, -10.69] | 1.514 | 0.0002 | 0.5804 |
| 4 | Intercept | 109.60 | 18.54 | [70.83, 148.4] | | **<0.0001** | |
| | Region [Asia] | −16.57 | 7.56 | [-32.39, -0.76] | 3.935 | **0.0409** | |
| | Region [Middle East] | −0.06 | 9.06 | [-19.01, 18.89] | 2.312 | 0.9950 | |
| | Region [Africa] | −12.87 | 12.02 | [-38.03, 12.29] | 1.472 | 0.2977 | |
| | Region [Latin America] | −11.15 | 8.84 | [-29.66, 7.35] | 2.817 | 0.2224 | |
| | BMI | −0.33 | 0.36 | [-1.08 to 0.41] | 1.578 | 0.3625 | |
| | Age | 0.23 | 0.26 | [-0.32, 0.78] | 2.747 | 0.3920 | |
| | T2DM [Yes] | −23.51 | 6.74 | [-37.61, -9.42] | 3.233 | **0.0024** | 0.5861 |

Multiple linear regression of FEV1% of predicted. Model 1 is Type 2 diabetes alone, model 2 is with the addition of BMI, model 3 with the addition of geographically region and model 4 with the addition of age. Analysis was conducted on never smokers without an established lung disease. The significant results of the best fit model are presented in bold. SE standard error; VIF variance inflation factor; CI confidence interval.

numbers (n = 1) to fully elucidate any possible significant differences. Furthermore, due to there only being one article that separated diabetes into lean, overweight and obese categories [78], it was not possible to perform a meta-analysis, heterogeneity analysis nor publication bias analysis. Finally, most of the included studies did not hold data for correlation between men and women, stratified smoking status, medication usage and socioeconomic status. While never smokers' data was used for the regression analysis, only one paper provided definitive groupings according to the diabetes medication liraglutide [115], no article could be split into educational status in line with our participant groupings and only one article provided information on occupational status [55]. However, similarly, this information could not be categorised according to the groupings for this investigation. Six papers provided male and female data that could be separated according to this article's groupings [36,40,62,64,76,92], but only two of these were Type 2 diabetes articles. Consequently, there was insufficient data to allow for analysis of these potential confounding variables.

## Conclusion

Type 2 diabetes is a significant independent contributor to pulmonary decline. In the absence of a known lung disease, hyperglycaemia is an independent risk factor for airway restriction but does not meet the clinical definition of a restrictive lung disease. Hyperglycaemia does not further exacerbate lung decline in asthma but reduces FVC% of predicted in COPD. This may be confounded by corticosteroid use. The higher prevalence of obstructive lung conditions in people with Type 2 diabetes may result from the action of inflammatory mediators on the airway. Therefore, it is proposed that early monitoring of lung function after a diagnosis of Type 2 diabetes is warranted to ensure early identification and treatment of respiratory decline.

**Table 3. Multiple linear regression of FVC% of predicted.**

| Model | Predictor | Estimate (β) | SE | 95% CI | VIF | P value | Goodness of Fit (R²) |
|---|---|---|---|---|---|---|---|
| 1 | Intercept | 92.17 | 3.03 | [85.98, 98.37] | | <0.0001 | |
| | T2DM [Yes] | −15.74 | 4.22 | [-24.36, -7.11] | 1.000 | 0.0008 | 0.3245 |
| 2 | Intercept | 99.76 | 12.17 | [74.38, 125.1] | | <0.0001 | |
| | BMI | −0.14 | 0.38 | [-0.93, 0.64] | 1.057 | 0.7081 | |
| | T2DM [Yes] | −18.88 | 5.36 | [-30.07, -7.70] | 1.057 | 0.0021 | 0.3865 |
| 3 | Intercept | 133.60 | 17.40 | [96.71, 170.5] | | **<0.0001** | |
| | Region [Asia] | −21.36 | 8.37 | [-39.09, -3.62] | 3.405 | **0.0213** | |
| | Region [Middle East] | −0.15 | 10.79 | [-23.03, 22.73] | 1.831 | 0.9891 | |
| | Region [Africa] | −17.86 | 13.23 | [-45.91, 10.18] | 1.441 | 0.1957 | |
| | Region [Latin America] | −14.52 | 9.58 | [-34.82, 5.78] | 2.608 | 0.149 | |
| | BMI | −0.69 | 0.39 | [-1.53, 0.14] | 1.553 | 0.0975 | |
| | T2DM [Yes] | −23.67 | 5.61 | [-35.57, -11.77] | 1.556 | **0.0007** | 0.6349 |
| 4 | Intercept | 139.8 | 29.23 | [77.16, 202.5] | | 0.0003 | |
| | Region [Asia] | −22.27 | 9.511 | [-42.67, -1.87] | 3.737 | 0.0345 | |
| | Region [Middle East] | −1.232 | 12.15 | [-27.29, 24.82] | 2.032 | 0.9207 | |
| | Region [Africa] | −18.97 | 14.65 | [-50.40, 12.46] | 1.553 | 0.2165 | |
| | Region [Latin America] | −16.02 | 11.6 | [-40.90, 8.87] | 3.336 | 0.1891 | |
| | BMI | −0.7128 | 0.4473 | [-1.67, 0.25] | 1.631 | 0.1333 | |
| | Age | −0.1222 | 0.4553 | [-1.10, 0.85] | 3.973 | 0.7923 | |
| | T2DM [Yes] | −21.9 | 9.632 | [-42.56, -1.25] | 3.833 | 0.0392 | 0.6257 |

Multiple linear regression of FVC% of predicted. Model 1 is Type 2 diabetes alone, model 2 is with the addition of BMI, model 3 with the addition of geographically region and model 4 with the addition of age. Analysis was conducted on never smokers without an established lung disease. The significant results of the best fit model are presented in bold. SE standard error; VIF variance inflation factor; CI confidence interval.

## Importance of the study

People with Type 2 diabetes have lower lung function and a higher risk of lung diseases including asthma and COPD than those who do not have diabetes. Yet, lung function is not routinely measured as a complication of Type 2 diabetes. This study provides evidence that Type 2 diabetes, independent of the confounding influence of obesity, significantly impairs lung function and proposes regular lung function monitoring may be beneficial in this population.

## Novelty statement

What is already known?

• Restrictive lung disease is associated with Type 2 diabetes. However, those with diabetes have high risk of obstructive lung diseases.

• Obesity is a major risk factor for both lung disease and Type 2 diabetes development.

What has this study found?

• Mild airway restriction is observed in Type 2 diabetes, but an independent role for obesity was not observed.

• Having co-morbid Type 2 diabetes does not lead to further worsening of lung function compared to individuals without diabetes when asthma is established.

- Type 2 diabetes further decreases the FVC% of predicted in people who develop COPD, potentially indicating a change to a restrictive pattern due to Type 2 diabetes.

What are the implications of the study?

- People with Type 2 diabetes should receive lung function monitoring to identify and treat respiratory decline.

## Supporting information

**S1 File. PICO (Population/Problem, Intervention/Exposure, Comparison/Control, Outcome) method used to formulate the research question.** Exclusion and inclusion criteria according to the PICO method to address the research question. *BMI* body mass index; *FEV1* forced expiratory volume in one second; *FVC* forced vital capacity; *COPD* chronic obstructive pulmonary disease; *GOLD* global initiative for chronic obstructive lung disease; *GINA* global initiative for asthma; *RCT* randomised controlled trial.
(DOCX)

**S2 File. Embase and Ovid MEDLINE search strategy with resulting article numbers for asthma and COPD.** Search strategies for COPD and Asthma in Ovid MEDLINE and Embase online databases with resulting article numbers before duplicates were removed. The initial COPD search (a, b) was carried out in November 2021 and the initial asthma search (c, d) in May 2022. The update checks for both asthma and COPD (e, f) were carried out in March 2023, with an additional check carried out in 2024 for articles up until January 2024 (g, h).
(DOCX)

**S3 File. Summary of acceptable disease definitions used in the final selected papers.** Complete list of acceptable definitions of disease diagnosis for COPD, Type 2 diabetes, obesity, and asthma. Confirmation was accepted if any one of the points within each category was present, except for 'clinical correlation' for COPD diagnosis, 'positive response to questionnaire asking has a medical person diagnosed you with Type 2 diabetes' for diabetes, and 'clinical symptoms' for asthma diagnosis, which needed extra forms of diagnosis for confirmation. *BMI* body mass index; *FEV1* forced expiratory volume in one second; *FVC* forced vital capacity; *COPD* chronic obstructive pulmonary disease; *HbA1c* glycated haemoglobin; *WHO* world health organisation; *GOLD* global initiative for chronic obstructive lung disease; *GINA* global initiative for asthma; *MBPT* methacholine bronchial provocation test; *PC20* provocative concentration of methacholine causing a 20% drop in FEV1; *ACQ* asthma control questionnaire.
(DOCX)

**S4 File. Adapted Newcastle-Ottawa Scale for cross-sectional studies.** An adapted Newcastle-Ottawa Scale (NOS) was used for cross-sectional studies. Two stars were given to properly recorded spirometry measurements in Outcomes that followed the American Thoracic Society (ATS) and/or European Respiratory Society (ERS) standard guidelines. As the main objective of this review is to obtain lung function measurements, one star was given when spirometry values were available but the nature of how the values were obtained was not clear or if the results were not recorded properly.
(DOCX)

**S5 File. Newcastle-Ottawa Scale for cohort studies.** An adapted Newcastle-Ottawa Scale (NOS) was used for cohort studies. Two stars were given to properly recorded spirometry measurements in Outcomes that followed the American Thoracic Society (ATS) and/or European Respiratory Society (ERS) standard guidelines. One star was given when spirometry values were available but the nature of how the values were obtained was not clear or if the results were not recorded properly.
(DOCX)

**S6 File. Articles assessed during the second screening.**
(PDF)

**S7 File. Quality assessment for cross-sectional studies (a), cohort studies (b), case-control studies (c), and Risk of Bias for interventional studies (d).** Quality assessment of cross-sectional studies (n = 74) using an adapted version of the Newcastle-Ottawa scale (See S3) (a). Studies were deemed as low, medium or high quality if they received 0–3, 4–5 or 6–7 stars, respectively. Quality assessment of cohort studies (n = 13) using the Newcastle-Ottawa scale for cohort studies (b). Studies were deemed as low, medium or high quality if they received 0–3, 4–7 or 8–10 stars, respectively. Quality assessment of case-control studies (n = 4) using the Newcastle-Ottawa scale for case-control studies (c). Studies were deemed as low, medium or high quality if they received 0–3, 4–6 or 7–9 stars, respectively. Risk of Bias for interventional studies (n = 2) using Cochrane Risk of Bias tool (RoB2) (d). Studies were rated as having either low, some concerns or high risk of bias.
(DOCX)

**S8 File. BMI, age, FEV1, FVC and FEV1/FVC ratio separated by BMI category.** Statistical significance between all study categories for age, BMI, FEV1% of predicted, FVC% of predicted and FEV1/FVC (L/L%) ratio is presented and separated by BMI status. Data presented as mean ± standard deviation. *BMI* body mass index; *FEV1* forced expiratory volume in one second; *FVC* forced vital capacity; *COPD* chronic obstructive pulmonary disease; *T2D* Type 2 diabetes. * $P \leq 0.05$, ** $P \leq 0.01$, *** $P \leq 0.001$, **** $P \leq 0.0001$. Lean BMI 18.5–24.9 kg/m$^2$, overweight BMI 25–29.9 kg/m$^2$, obese BMI ≥ 30 kg/m$^2$.
(DOCX)

**S9 File. PRISMA 2020 Checklist.**
(DOCX)

**S10 File. PRISMA 2020 for Abstracts Checklist.**
(DOCX)

**S11 File. Full quality assessment and risk of bias results.**
(DOCX)

**S12 File. Multiple linear regression of FEV1/FVC.** Multiple linear regression of FEV1/FVC. Model 1 is Type 2 diabetes alone, model 2 is with the addition of BMI, model 3 with the addition of geographically region and model 4 with the addition of age. Analysis was conducted on never smokers without an established lung disease. None of the models were normally distributed. SE standard error; VIF variance inflation factor; CI confidence interval.
(DOCX)

## Author contributions

**Conceptualization:** Sarah Atkinson, Paula L. McClean, Catriona Kelly.

**Data curation:** Roanne Lecky, Svitlana Grogan, Catriona Kelly.

**Formal analysis:** Roanne Lecky, Svitlana Grogan, Priyank Shukla, Sarah Atkinson, Paula L. McClean, Catriona Kelly.

**Funding acquisition:** Priyank Shukla, Paula L. McClean, Catriona Kelly.

**Investigation:** Roanne Lecky, Svitlana Grogan.

**Methodology:** Roanne Lecky, Sarah Atkinson, Paula L. McClean, Catriona Kelly.

**Project administration:** Priyank Shukla.

**Supervision:** Priyank Shukla, Paula L. McClean, Catriona Kelly.

**Writing – original draft:** Roanne Lecky, Svitlana Grogan, Sarah Atkinson, Paula L. McClean, Catriona Kelly.

**Writing – review & editing:** Roanne Lecky, Priyank Shukla, Paula L. McClean, Catriona Kelly.

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
