## [Decision Letter · Decision Letter 0]

16 May 2025

Dear Dr. Kelly,

Thank you for submitting your manuscript to PLOS ONE. After careful consideration, we feel that it has merit but does not fully meet PLOS ONE’s publication criteria as it currently stands. Therefore, we invite you to submit a revised version of the manuscript that addresses the points raised during the review process.

We look forward to receiving your revised manuscript.

Kind regards,

Hidetaka Hamasaki

Academic Editor

PLOS ONE

2. As required by our policy on Data Availability, please ensure your manuscript or supplementary information includes the following:

[R.L. and S.G are supported by PhD Studentships from the Department for the Economy, Northern Ireland.].

[R.L. and S.G are supported by PhD Studentships from the Department for the Economy, Northern Ireland.]

[R.L. and S.G are supported by PhD Studentships from the Department for the Economy, Northern Ireland.].

5. In the online submission form, you indicated that [The data underlying the results presented in the study are available from the corresponding author upon request (Dr Catriona Kelly, c.kelly@ulster.ac.uk).here].

Reviewers' comments:

Reviewer's Responses to Questions

**Comments to the Author**

1. Is the manuscript technically sound, and do the data support the conclusions?

Reviewer #1: Partly

Reviewer #2: Yes

2. Has the statistical analysis been performed appropriately and rigorously?

Reviewer #1: N/A

Reviewer #2: Yes

3. Have the authors made all data underlying the findings in their manuscript fully available?

Reviewer #1: No

Reviewer #2: Yes

4. Is the manuscript presented in an intelligible fashion and written in standard English?

Reviewer #1: No

Reviewer #2: Yes

Reviewer #1: The manuscript is technically correct, methodologically rigorous, and uses correct statistical analyses.

No meta-analysis, cross-sectional data, and confounders left unexamined limit the strength of the inferences.

Most conclusions are backed by the data, but more work is required to validate some claims (like that obesity plays a role in lung function).

The manuscript could be improved (meta-analysis, confounding variables, and causal interpretations) if the authors clarify the limitations.

Outcome consistency is affected by heterogeneity among included studies (e.g., different methodologies, diagnostic criteria, and population characteristics). The reliance on cross-sectional studies (87 of 109 studies) precludes an established causal relationship.

This publishing goes against the findings from several previous studies showing how obesity is harmful with regard to lung function decline. The authors recognize this, but are not sufficiently clear about what might explain the difference.

The analysis may not fully adjust for confounding factors (e.g., smoking, medication use, disease duration). The trial does not descriptively assess for publication bias that may affect the reported findings.

Reviewer #2: The manuscript entitled “The role of obesity and Type 2 diabetes in lung health: A systematic review” focuses on the study of comorbidities, seeking to differentiate the role of different variables that the literature has indicated are associated. This systematic review concludes that type 2 diabetes, independently of the influence of obesity, significantly impairs lung function and proposes that periodic monitoring of lung function may be beneficial in this population.

This reviewer considers that the authors performed a careful review of the literature that justifies this research and followed a very careful methodology considering the PRISMA guidelines. Finally, the results and discussion are written in congruence with the objectives of the study.

Only two observations are made:

1. Review the first line of the introduction to ensure the wording “...Type 2 diabetes is due to microvascular and microvascular complications” is correct.

2. In the results section, subtitled “Association of Type 2 diabetes and/or obesity with lung function: Review of the literature”, it is mentioned “Initial analysis was performed on all studies...”. Please clarify the discrepancy between the total number of articles to be analyzed (109) and the breakdown of articles in this section (170).

**Do you want your identity to be public for this peer review?** For information about this choice, including consent withdrawal, please see our Privacy Policy

Reviewer #1: **Yes:** Anees Alyafei

Reviewer #2: No

---

## [Author Response · Author response to Decision Letter 1]

18 Nov 2025

Please see attached response to reviewers document

---

## [Editor Report · Decision Letter 1]

25 Dec 2025

The role of obesity and Type 2 diabetes in lung health: A systematic review

PONE-D-25-07836R1

Dear Dr. Kelly,

We’re pleased to inform you that your manuscript has been judged scientifically suitable for publication and will be formally accepted for publication once it meets all outstanding technical requirements.

Kind regards,

Hidetaka Hamasaki

Academic Editor

PLOS One

Additional Editor Comments (optional):

Thank you for submitting the revised manuscript.

I apologize for the delay in reaching a publication decision. As we did not receive a response from Reviewer 1, I carefully evaluated your replies to the reviewers myself. Many of your responses were incorporated as study limitations, and therefore they do not fully address all of the reviewers’ requests. However, given the inherent limitations related to the study design, these issues can only be discussed as limitations.

Overall, the manuscript is scientifically sound, and I am pleased to inform you that it has been accepted for publication. We hope that future studies will address the issue of heterogeneity and enable the conduct of a meta analysis.
---

## [Editor Report · Acceptance letter]

PONE-D-25-07836R1

PLOS One

Dear Dr. Kelly,

I'm pleased to inform you that your manuscript has been deemed suitable for publication in PLOS One. Congratulations! Your manuscript is now being handed over to our production team.

Kind regards,

on behalf of

Dr. Hidetaka Hamasaki

Academic Editor

PLOS One